# *Lavandula x intermedia*—A Bastard Lavender or a Plant of Many Values? Part II. Biological Activities and Applications of Lavandin

**DOI:** 10.3390/molecules28072986

**Published:** 2023-03-27

**Authors:** Katarzyna Pokajewicz, Marta Czarniecka-Wiera, Agnieszka Krajewska, Ewa Maciejczyk, Piotr P. Wieczorek

**Affiliations:** 1Institute of Chemistry, University of Opole, 45-052 Opole, Poland; 2Institute of Biology, University of Opole, Oleska 22, 45-052 Opole, Poland; 3Department of Biotechnology and Food Science, Lodz University of Technology, 90-530 Lodz, Poland

**Keywords:** *Lavandula x intermedia*, *Lavandula hybrida*, *Lavandula angustifolia*, essential oil, antimicrobial activity, antioxidant activity, biocidal activity, anxiolytic activity, anti-inflammatory

## Abstract

This review article is the second in a series aimed at providing an in-depth overview of *Lavandula x intermedia* (lavandin). In part I, the biology and chemistry of lavandin were addressed. In part II, the focus is on the functional properties of lavandin and its applications in industry and daily life. While reviewing the biological properties, only original research articles employing lavandin were considered. Lavandin essential oil has been found to have antioxidant and biocidal activity (antimicrobial, nematicidal, antiprotozoal, insecticidal, and allelopathic), as well as other potential therapeutic effects such as anxiolytic, neuroprotective, improving sleep quality, antithrombotic, anti-inflammatory, and analgesic. Other lavandin preparations have been investigated to a much lesser extent. The research is either limited or inconsistent across all studies, and further evidence is needed to support these properties. Unlike its parent species—*Lavandula angustifolia* (LA)—lavandin essential oil is not officially recognized as a medicinal raw material in European Pharmacopeia. However, whenever compared to LA in shared studies, it has shown similar effects (or even more pronounced in the case of biocidal activities). This suggests that lavandin has similar potential for use in medicine.

## 1. Introduction

*Lavandula x intermedia* Emeric ex Loisel (LI), also known as lavandin, Dutch lavender, or bastard lavender, is a widely cultivated aromatic plant belonging to the family *Lamiaceae* Lindl. It is a hybrid of true lavender—*Lavandula angustifolia* (LA)—and spike lavender—*Lavandula latifolia* (LL). While it shares many similarities with its parent species, lavandin possesses unique characteristics that set it apart. This review is a continuation of an article entitled “*Lavandula x intermedia*—A Bastard Lavender or A Plant of Many Values? Part I. Biology and Chemical Composition of Lavandin” [1]. Part I covered the biological and chemical characteristics of *L. x intermedia*, including taxonomy, geographical range, morphological features, popular cultivars, cultivation, and essential oil production. Additionally, the chemical composition of its essential oil and hydrolate was thoroughly discussed and compared to the parent species, taking into account the current industry standards such as ISO, European Pharmacopeia (Ph. Eur.), and WHO monographs. We stated that lavandin essential oil (further referred to as lavandin oil) has a similar chemical composition to LA, but with a higher concentration of terpenes that give it a camphor scent, making it less appealing for use in the perfume industry. However, LI has some benefits, such as a higher yield of essential oil and lower production cost, making it a favored lavender crop for farming. Nonetheless, despite its commercial success and widespread cultivation, there is a shortage of scientific research on the subject. The scientific community tends to focus on LA, a raw material recognized by European Pharmacopeia.

Furthermore, lavandin is often seen as an inferior—bastard lavender plant than true lavender. This assertion, however, aside from its use in the perfume industry, is not supported by reliable arguments. This raises the question of whether lavandin and lavandin-related products are not as valuable as those of true lavender in other non-perfumery applications. In Part II, we discuss all reported biological effects of *L. x intermedia* essential oil and its other extracts in an attempt to answer this question. Furthermore, we reviewed the current applications of lavandin in industries and everyday life.

As we embarked on this review, we aimed to thoroughly examine all available original scientific research articles on LI and explore its potential as an alternative to LA in various applications. There is a need for a review article dedicated to lavandin, as no such article has been published thus far, let alone one that explores its biological activities. Through this review, we hope to contribute to the understanding of this plant and lead to a greater appreciation of its importance in the scientific community and, consequently, its inclusion in more scientific studies alongside *Lavandula angustifolia*.

## 2. Biological Activities of Lavandin

The most obvious and apparent biological property of lavandin is its smell. It is caused by the volatile chemicals, mainly oxygenated monoterpenes, secreted and stored in the aerial parts of the plant [2,3,4,5]. Most of the applications of LI in industries and daily life result from this significant feature of this plant. Apart from its smell, lavandin, like many other aromatic herbs, is associated with numerous biological effects. This section aims to review the current state of knowledge on the biological activities of *L. x intermedia*, either its essential oil or any other kind of extract. We have reviewed all research articles we could find on the subject, including all Scopus hits for the phrases such as “Lavandula and intermedia” and “Lavandula and hybrida”. We did not consider or cite any articles that made generalizations about the biological effect of lavandin based on studies made on the other taxa within the *Lavandula* genus. This practice appears to be common in the scientific literature.

### 2.1. Biocidal Activities

A detailed examination of the scientific literature regarding the biological activities of LI allows for a conclusion that most research conducted in this field is related to the biocidal properties of lavandin, specifically its essential oil, with hydrolates and other plant extracts being studied sporadically. Table 1 and Table 2 summarize all of the original literature in this respect.

#### 2.1.1. Antimicrobial

Many essential oils (EOs) exhibit antimicrobial properties. They have been used for centuries in traditional medicine and for embalming a corpse. Even though multiple EOs have demonstrated antimicrobial action, only some possess the potential to be used as antimicrobial agents. The real-world effect is usually significantly weaker compared to antibiotics and other synthetic compounds [6]. *L. angustifolia* has been proven to be effective against many bacteria, fungi, and some viruses [6,7,8,9]. There is also multiple evidence for the antibacterial and antifungal action of lavandin oil, but according to our best knowledge—there is not any research investigating its antiviral effect.

Antimicrobial studies of lavandin oil, like other essential oils, are usually conducted in vitro with the use of agar diffusion (disc or well) methods and/or dilution methods. The diffusion methods, especially the disc diffusion method, are mainly used for antimicrobial susceptibility testing. Dilution methods are the most suitable for the determination of minimum inhibitory concentration (MIC), minimal lethal concentration (MLC), minimum bactericidal concentration (MBC), and minimum fungicidal concentration (MFC) values due to the fact they enable the calculation of the concentration of the tested antimicrobial chemical in the broth or agar media [7,10,11]. The review of all antimicrobial activity studies of lavandin preparations (mostly essential oils) for both bacteria and fungi reported in the literature is presented in Table 1.

The antibacterial and antifungal effect of lavandin EOs against many gram-positive and negative bacteria was demonstrated by multiple researchers (cited in Table 1). Different lavandin cultivars were tested. For example, Garzoli and coworkers tested EO of the very popular cultivar Grosso grown in Italy against *Escherichia coli*, *Acinetobacter bohemicus*, *Pseudomonas fluorescens*, *Bacillus cereus*, and *Kocuria marina* and found bactericidal effect on Gram-negative bacteria and a bacteriostatic effect on Gram-positive bacteria both for the liquid and vapor phases [12]. According to the various tests that authors conducted, *A. bohemicus* was the most vulnerable strain to lavandin essential oil. It exhibited an inhibition zone of 47 mm (greater than of positive control gentamicin) and had MIC of 0.47% in the broth microdilution test. *P. fluorescens* was the most resistant among all the strains tested. It had an inhibition zone of just 8.5 mm and MIC of 3.75%. Bajalan et al. found the high antibacterial activity of Iranian lavandin oil from leaves against G− *E. coli* and G+ *Streptococcus agalactiae* and moderate against G− *K. pneumoniae* and *S. aureus* [13]. The antibacterial effect in vitro and in vivo in mice against *Citrobacter rodentium* (G−) was also indicated by Baker et al. [14]. When *L. x intermedia* and *L. angustifolia* are considered in one study, usually lavandin oil possesses similar or stronger antibacterial and antifungal effects than true lavender oil. Jianu et al. investigated Romanian LI and LA essential oil against *Enterococcus faecium*, *Shigella flexneri*, *Salmonella typhimurium*, *Escherichia coli*, and *Streptococcus pyogenes*. The studied oils presented significant bactericidal effects against *S. flexneri*, *S. aureus*, and *E. coli* but not against *S. pyogenes*. In most cases, *L. x intermedia* antibacterial activity was higher [15]. Stronger action of LI was also observed by Tardugno and coworkers, who tested EOs of different cultivars of LI and LA (Italian origin) against *Listeria monocytogenes* [16]. Di Vito et al. indicated similar antibacterial and antifungal properties of both lavender oils with a slightly higher effect for *L. intermedia* [17]. On the other hand, Robu et al. tested Romanian LI and LA essential oils against *S. aureus*, *S. pyogenes*, *P. aeruginosa*, *E. coli*, and *Candida albicans*, and they noticed that *L. angustifolia* essential oil was more active on certain bacterial strains, but *L. x intermedia* EO was more effective against *Candida* [18]. Antifungal properties of LI EO in high doses against *Candida albicans* were noticed by Karakaş and Bekler [19]. Moon et al. have also observed the antifungal activity of oils of lavandin and other species of lavender they studied. The EOs of three different cultivars of LI and LA oils were effective against *Aspergillus nidulans* and *Trichophyton mentagrophytes* [20]. Lavandin oil was also proved by Larrán et al. to be fungistatic against some strains of studied *Ascosphaera apis*—the fungus causing the chalkbrood disease of bees [21]. However, Erland and coworkers tested LI ‘Provence’ and ‘Grosso’ and LA oils and observed no significant antifungal effect against three agricultural pathogens, except some activity of LI ‘Provence’ oil against *B. cinerea* [22].

When comparing the antimicrobial activity of lavandin or true lavender oil with the oils of other aromatic plants, it has been found that some plants are far more effective, usually due to their high content of phenolic compounds, which are characterized by strong antimicrobial properties. Tardugno et al. conducted in vitro screening to assess the antimicrobial activity of 14 essential oils against oral pathogenic bacteria. It was indicated that lavandin oils showed moderate activity among all tested oils, with MIC ranging from 2–512 μL/mL. The most effective oils were those derived from *Thymus vulgaris* and *Rosmarinus officinalis*, which had MICs of 4–16 and 1–32, respectively [23]. Rota and coworkers studied the antimicrobial activity against selected foodborne pathogenic bacteria. Once again, lavandin oil showed intermediate antibacterial activity among the tested samples. As expected, the biggest effects were observed for *T. vulgaris* and *Satureja montana* oils, whereas the weakest effects were noticed for *Salvia sclarea* and *Hyssopus officinalis* [24]. The above-mentioned Di Vito et al. also studied other than lavender EOs and found that both lavandin and true lavender oils exhibited weaker activity against tested microorganisms (bacteria, drug-resistant yeasts, and fungal dermatophytes) than oils containing a lot of thymol and/or carvacrol, such as those from *Origanum hirthum*, *S. montana*, *Monarda didyma*, and *Monarda fistulosa*. The same authors also demonstrated that essential oils work much stronger than hydrolates, which exhibited mostly high MIC and MLC values (above 50%), whereas lavender essential oils had MIC and MLC values mostly above 2% [17]. No antibacterial activity of *Lavandula* spp. hydrolates was observed by Moon et al. The authors also evaluated aqueous and ethanolic extracts and found that water extract had no activity, while some ethanolic extracts were effective against *Proteus vulgaris* [20]. Ramić and colleagues tested lavandin essential oil and ethanolic extracts and observed strong antibacterial activity against one of the major food-borne pathogens—*Campylobacter jejuni*, with EOs exhibiting the strongest effect and MIC of 0.25 mg/mL, whereas ethanolic extracts had MIC of 0.5–1 mg/mL) [25]. The antimicrobial activity of lavandin ethanolic extracts of the same ‘Budrovka’ cultivar was also confirmed by other researchers—Blazenkovic et al., who found that ethanolic extracts, especially those from flowers, exhibited antimicrobial activity against a broad spectrum of bacteria, yeasts, molds, and dermatophytes. The antimicrobial activity of the extracts decreased in the order of plant part: flowers > leaves > inflorescence stalks [26].


molecules-28-02986-t001_Table 1Table 1The antimicrobial activities of lavandin natural products that are reported in the literature. Only original research articles were considered.Targeted OrganismStudied AgentOutcomeCitationAntibacterial effect*Citrobacter rodentium*—a bacteria used to model infections by the human-specific enteric bacterial pathogens*L. x intermedia* ‘Okanagan’ and wild-type EOsAntimicrobial activity against the pathogen was observed in vitro and in vivo in mice. ‘Okanagan’ EO (OEO, a cultivar rich in 1,8-cineole and borneol) exhibited more potent antibacterial activity than EO from wild-type lavandin. OEO inhibited systemic infection of *C. rodentium* in mice and modulated the enteric microbiota. Firmicutes enteric bacteria, segmented filamentous bacteria, *Clostridia* spp., and *Eubacterium rectale* were significantly increased in the ceca, while several other microbes such as *Bacillus* spp., *Lactobacillus* spp., and the *Clostridium coccoides* group, remained the same. Both EOs inhibited adherence (through the action of 1,8-cineole and borneol) and growth of *C. rodentium* in vitro. In disk diffusion assays, 20 μL OEO showed a 24 mm inhibition zone, and wild-type EO 22 mm (with a statistically insignificant difference). 1,8-cineole was the oil constituent with antimicrobial activity, while no camphor and borneol activity was found. However, in isolation, this monoterpenoid was not as effective as in oils suggesting that a combination of constituents in EO synergistically produces the greatest antimicrobial activity.[14]*Enterococcus faecium*, *Shigella flexneri*, *Salmonella typhimurium*, *Escherichia coli*,*Streptococcus pyogenes**L. x intermedia* and *L. angustifolia* EOs, grown in western RomaniaStudied lavender and lavandin EOs presented significant bactericidal effects against *S. flexneri* (inhibition zones for 20 μL EO of LA—20 mm, LI—26 mm), *S. aureus* (LA—20 mm, LI—20 mm), and *E. coli* (LA—20 mm, LI—21 mm), but not against *S. pyogenes* in disc diffusion tests. In most cases and doses, *L. x intermedia* activity was higher. However, the studied EOs, although distilled from flowering shoots, had an unusual composition, with almost none of the linalool and its acetate.[15]*Escherichia coli*, *Acinetobacter bohemicus*, *Pseudomonas fluorescens*, *Bacillus cereus*, *Kocuria marina**L. x intermedia* EO ‘Grosso’, grown in ItalyA bactericidal effect on G− bacteria and a bacteriostatic effect on G+ bacteria were noticed both for the liquid and vapor phases. Below are given the MIC and MBC values for LI _EO_ and positive control gentamicin (_GEN_):*P. fluorescens* MIC_EO_ = 3.75%, MIC_GEN_ = 1.56 µg/mL,MBC_EO_ = 7.51%, MBC_GEN_ = 6.25%;*E. coli* MIC_EO_ = 1.87%, MIC_GEN_ = 3.12 µg/mL,MBC_EO_ = 1.87%, MBC_GEN_ = 6.25%;*K. marina* MIC_EO_ = 1.87%, MIC_GEN_ = 0.39 µg/mL,bacteriostatic, MBC_GEN_ = 1.56%;*B. cereus* MIC_EO_ = 0.94%, MIC_GEN_ = 1.56 µg/mL,bacteriostatic, MBC_GEN_ = 3.12%;*A. bohemicus* MIC_EO_ = 0.47%, MIC_GEN_ = 0.08 µg/mL,MBC_EO_ = 0.47%, MBC_GEN_ = 0.31%.The findings from various assays indicated that *A. bohemicus* was the most susceptible strain to lavandin EO (10 μL), displaying an inhibition zone of 47 mm and MIC of 0.47%. *B. cereus* was the second most sensitive microorganism, exhibiting an inhibition zone of 21.5 mm and MIC of 0.94%. *E. coli* and *K. marina* showed similar results, with inhibition zones of 13.0 and 14.5 mm, respectively, and both exhibited MIC of 1.87% in the microdilution test. *P. fluorescens* was the most resistant bacteria among all the tested strains, showing an inhibition zone of only 8.5 mm and MIC of 3.75%[12]*Listeria monocytogenes* and *Salmonella enterica* strains isolated from food products*L. x intermedia* EO ‘Abrial’, ‘Alba’, ‘Rinaldi Ceroni’, ‘Sumiens’; additionally, *Lavandula angustifolia* EO tested, grown in ItalyAll tested EOs showed antimicrobial effects against *L. monocytogenes* strains. Specifically, lavandin cultivars ‘Abrial’, ‘Rinaldi Ceroni’, and ‘Sumiens’, which have high levels of linalool, camphor, and 1,8-cineole, were highly effective against *L. monocytogenes*—particularly against strains obtained from the clinical environment, with MICs ranging from 0.3 μL/mL up to 5 μL/mL. A weaker effect of EOs was observed against *S. enterica*, with MICs ranging from 10 to 80 μL/mL. Among the plants tested, LI cultivar ‘Rinaldi Ceroni’ was the most active with MICs 10–40 μL/mL. The essential oil of *L. angustifolia* showed comparable or weaker antimicrobial activity compared to the most potent LI cultivar, with MICs of 1.5–5 μL/mL for *L. monocytogenes* and 40–80 μL/mL for *S. enterica*.[16]*S. agalactiae*, *S. aureus*, *E. coli*, *K. pneumoniae**L. x intermedia* EOs of Iranian origin made from leaves, no cultivar specifiedIn disc diffusion tests, LI essential oils showed significant antibacterial activity against *E. coli* and *S. agalactiae*, with inhibition zones of 15–23 mm and 12–17 mm, respectively, when a 20 μL amount was used. Slightly weaker effects were observed against *K. pneumoniae* (9–16 mm) and *S. aureus* (9–15 mm). The studied EOs were obtained from leaves and had a different composition compared to the standard EO distilled from flowering tops. In particular, it had almost no linalool and its acetate and consisted mostly of 1,8-cineole, borneol, and camphor. The authors observed that antibacterial activity is significantly correlated with the presence of 1,8-cineole.[13]*E. coli*, *S. aureus*,*B. cereus**L. x intermedia* EO ‘Super’The study evaluated the antimicrobial effects of both free and encapsulated lavandin essential oil against three pathogenic bacteria. Before analyzing the formulations with encapsulated essential oil, the antibacterial activity of pure lavandin essential oil was tested in vitro, and MICs were 7.1 mg/mL for both *E. coli* and *S. aureus* and 3.6 mg/mL for *B. cereus*. Lavandin oil’s antibacterial activity could be enhanced by encapsulation due to the protection and controlled release of the essential oil. Soybean lecithin was found to be an efficient carrier material for LI essential oil, with better results than other carriers, and it was suggested that liposomes formed could cross both phospholipid layers of Gram-negative bacteria and deliver the essential oil inside the cell of bacteria.[27]*Campylobacter jejuni* biofilms on abiotic surfaces*L. x intermedia* ‘Bila’, ‘Budrovka SN’, ‘Budrovka’; EOs and ethanolic extracts, grown in CroatiaAll studied EOs possessed the best antibacterial activity with a minimal inhibitory concentration of 0.25 mg/mL. A weaker effect was observed for ethanolic extracts of flowers before distillation and post-distillation with MIC of 0.5–1 mg/mL. Lavandin ethanolic extracts of flowers prior to distillation were found to be more efficient in decreasing intercellular signaling and adhesion of *C. jejuni* as compared to lavandin EOs and ethanolic extracts of post-distillation waste material. However, lavandin EOs exhibited a slightly stronger impact on inhibiting the formation of biofilm. The authors concluded that lavandin formulations can be used as antimicrobial agents to control the development of *C. jejuni* biofilm.[25]*Lactobacillus* spp.,*S. mutans* (oral pathogenic bacteria)*L. x intermedia* EO ‘Grosso’, ‘Sumian’; EOs of other plants additionally testedLavandin EOs exhibited an antibacterial effect. A microwell dilution assay revealed MIC for EO of ‘Grosso’ as 16 μL/mL for two strains of *Lactobacillus* spp. and 16 and 256 μL/mL for two strains of *S. mutans*. ‘Sumian’ EO had MICs of 32 and 2 μL/mL for *Lactobacillus* strains and 32 and 512 μL/mL for *S. mutans*. Lavandin oils had a weaker effect than *Thymus* and *Rosmarinus* EOs and their MICs were 4–16 and 1–32 μL/mL, respectively. The authors found that the antibacterial power was positively correlated with the content of menthol, thymol, and carvacrol in the essential oil. The effects of LI oils were not significant enough to include them in the further stages of the experiment, which involved screening combinations of essential oils and chlorhexidine.[23]*S. pyogenes*, *S. aureus* (MRSA), *Citrobacter freundii*, *Proteus vulgaris*, *E. coli*, VRE, *Propionibacterium acnes*, *Pseudomonas aeruginosa**L. x intermedia* ‘Grosso’’ ‘Seal’, ‘Miss Donnington and other *Lavandula* spp. essential oils, hydrolates, and aqueous and ethanolic foliage extractsThe lavender hydrosols and aqueous foliage extracts did not possess antibacterial activity in the disc diffusion assay (10 μL of agent). Some ethanolic extracts displayed activity against *P. vulgaris* (inhibition zones of 7–9 mm) but no action against other studied microorganisms.EOs exhibited antibacterial effects (inhibition zones of 7–15.5 mm). There was considerable variability in the activity of the essential oils. However, no oil presented the highest antibacterial activity against all bacteria. Furthermore, there was no observed correlation between the content of major chemical components and antibacterial activity. *P. aeruginosa* was the only bacterium not susceptible to any studied essential oil of Australian origin[28]*Paenibacillus larvae*—American Foulbrood Disease of honeybees’ pathogen*L. x intermedia* and other plants EOsThe antibacterial activity of lavandin oil against eight strains of *P. larvae* was demonstrated, with MIC ranging from 0.45 to 0.6 μL/mL. Though, the stronger activity of some other EOs, such as *Cymbopogon citratus* (0.05–0.1), *Origanum vulgare* (0.25–0.45), *Satureja hortensis* (0.2–0.25), and *Thymus vulgaris* (0.1–0.15) was observed. On the other hand, a weaker effect was observed for EOs of *Eucalyptus globulus* (>0.7), *Mentha x piperita* (0.6–0.65), and *Rosmarinus officinalis* (0.7). The authors suggested that lemon grass and thyme essential oils have the potential to inhibit Foulbrood Disease in honeybee colonies.[29]Antibacterial and antifungal effects*S. aureus*, *Streptococcus pyogenes*, *P. aeruginosa*, *E. coli*, *Candida albicans**L. x intermedia* and *L. angustifolia* EOs, grown in RomaniaThe study evaluated the antimicrobial activity of essential oils against four pathogenic bacteria and one genus of yeasts that causes fungal infections. *L. angustifolia* EO was noticed to be more active against target bacterial strains compared to *L. x intermedia*. The calculated MIC value for LI was 0.5 µL/mL, and the MBC value was 1.00 µL/mL, whereas for LA, the MIC was 0.25 µL/mL and MBC was 0.25 µL/mL. On the other hand, the antifungal activity of LI essential oil was three times more effective, and 10 μL of LI oil produced a 17 mm zone of inhibition in the disc diffusion method compared to the 5 mm zone produced by LA oil.[18]31 strains of bacteria, yeasts, dermatophytes, and molds*L. x intermedia* ‘Budrovka’—ethanolic extracts, grown in CroatiaThe study presents the antibacterial and antifungal activity of *L. x intermedia* against 31 strains of bacteria, yeasts, dermatophytes, and molds. Antimicrobial activity of the extracts decreased in the order of flower > leaf > inflorescence stalk. The flower extract was effective against all tested bacteria (MIC: 0.1–1.5%; MBC: 0.2–2.5%). The leaf extract showed a weaker antibacterial effect (MIC: 1.25–10.0%; MBC: 2.5–12.5%), while the extract of stalks was the least effective (MIC: 2.5–12.5%; MBC: 5.0–15.0%). Ethanolic flower extracts had the strongest antifungal effect (MIC: 0.05–2.5%; MFC: 0.1–5.0%) compared to leaf and stalk extracts (leaf: MIC: 2–30%, MFC: 4–35%; stalk: MIC: 5–37,5%, MFC: 7.5–40%). Leaf and stalk extract showed little or no antibacterial action against several gram-negative bacteria and some fungi.[26]Bacteria: *S. aureus* (MRSA and MSSA), *S. pyogenes*, *Enterococcus faecalis* (VRE), and*Enterococcus faecium*. Drug-resistant yeasts: *C. albicans*, *C. parapsilosis*, *C. glabrata*, and *C. tropicalis*.Dermatophytes: *Trichophyton* spp. (*soudanense*, *tonsurans*, *rubrum*, *violaceum*), and *Microsporum canis*.*L. x intermedia* and other plants EOs and hydrolates, Italian originAntibacterial and antifungal activity of both *L. x intermedia* and *L. angustifolia* was shown. MIC and MLC of both lavender oils were mostly above 2% (*v*/*v*). The exception was *T. violaceum*, where the MIC and MLC did not exceed 0.2%. Accordingly, the antimicrobial activity of both lavender EOs was lower than that of *Origanum hirthum*, *Satureja montana*, *Monarda didyma*, and *M. fistulosa*. Hydrolates were much weaker, with MICs and MLCs mostly above 50%. LI oil was slightly more active than LA oil against bacteria and fungi.[17]*S. aureus*, *E. coli*, *C. albicans**L. x intermedia* and other plants EOs from leaves of Turkish origin
The authors studied the antimicrobial activities of EOs of *L. x intermedia* and other plants using the disk diffusion method. The effect was assessed based on the zones of inhibition (ZI). The antimicrobial effect varied depending on the microorganism type and the dose of the essential oil. The LI EOs exhibited potent effects against *C. albicans* at doses of 15 μL (resulting in a 30 mm zone of inhibition) and 7.5 μL (20 mm ZI) but moderate effects at lower doses of 3 μL (12 mm ZI) and 1.5 μL (6 mm ZI). The antimicrobial effect was weaker in the case of *E. coli* (7–12 mm ZIs) and *S. aureus* (6–13 mm ZIs) in all tested doses. Among all analyzed plants, *Thymbra spicata* and *Satureja macrantha* essential oils had stronger antimicrobial activity against all tested microorganisms.
[19]Antifungal effect*Cladobotryum mycophilum*—cobweb disease of button mushroom (*Agaricus bisporus*)*L. x intermedia* EO and other plants’ EOsIn vitro assays showed that the EOs obtained from *Thymus vulgaris* (median effective concentration EC_50_ of 35.5 mg/L) and *Satureja montana* (42.8 mg/L) were the most effective for inhibiting the mycelial growth of *C. mycophilum* and were also the most selective between *C. mycophilum* and *A. bisporus*. The EOs of *L. x intermedia* (EC_50_ = 146.6 mg/L) and *Thymus mastichina* (175.7 mg/L) had the strongest antifungal effect against *A. bisporus*, which could be attributed to the content of alcoholic monoterpenoids.[30]Agricultural pathogens: *Mucor piriformis*, *Botrytis cinerea*, *Penicillium expansum**L. x intermedia* ‘Grosso’ and ‘Provence’, and *L. angustifolia* EOsThe antifungal effect of essential oils from LA and LI, as well as individual essential oil constituents, was evaluated using disk diffusion assays against three agricultural pathogens. True lavender oil exhibited no significant effect. However, the LI ‘Provence’ oil showed some antifungal activity against *B. cinerea*, resulting in a zone of inhibition 7.5 mm at a concentration of 50 μL/mL of oil. Among the terpenes studied, carvacrol was the most effective component (although this component is not present in *Lavandula* oils), which showed an inhibition zone of 40 mm at a concentration of 50 μL/mL after 24 h. Lavandulol and linalool produced zones of inhibition of 9 and 7–8 mm, respectively.[22]*Aspergillus nidulans*, *Trichophyton mentagrophytes*, *Leptosphaeria maculans* and *Sclerotinia sclerotiorum*Australian-grown *L. x intermedia* ‘Grosso’, ‘Miss Donnington’, *L. angustifolia*, and other *Lavandula* species EOs and hydrolates All tested essential oils displayed some antifungal activity. EOs derived from LA, and all three tested LI cultivars caused a decrease in the growth of *A. nidulans* and *T. mentagrophytes* by at least 50% after 6 days of the experiment. *Lavandula stoechas* was particularly effective against the two agricultural fungi: *L. maculans* and *S. sclerotiorum*. No evidence of antifungal activity was observed for all hydrolates studied.[20]*Trichophyton mentagrophytes* (tinea pedis causing pathogen)*Lavandula x intermedia* and other plants’ ethyl acetate extractsThe results showed that most of the herbs exhibited potent vapor activity against *T. mentagrophytes*, of which Roman chamomile, curry plant, hyssop, lavandin, marjoram sweet, orange mint, spearmint, monarda, oregano, rosemary, rue sage, tansy, tarragon, thyme common and yarrow showed the most potent activity. The vapors of these herbs exhibited lethal properties by killing over 99.99% of fungus. The lavandin extract exhibited a strong antifungal effect producing a 36 mm zone of inhibition in agar diffusion assay.[31]*Ascosphaera apis*—a chalkbrood disease of honeybees*L. x intermedia* and other plants’ EOsLavandin oil was effective at 700 μL/L against 3 of 5 studied *A. apis* strains regardless of the geographical significance of the strains. At all concentrations tested, coriander oil was the most effective fungistatic control.[21]


Summarizing the above findings, there is no doubt that lavandin preparations, such as essential oil or ethanolic extract, possess antimicrobial activity. This activity is at least as strong as the activity of *L. angustifolia* or stronger. Hydrolates and other lavandin preparations do not exhibit antimicrobial power, or it is significantly weaker. Regarding the antimicrobial activity of essential oils, even though it is proven and well-established, the therapeutic effect is significantly weaker when compared to synthetic antibiotics. Essential oils are volatile, and their ability for quick vaporization can shorten their effectiveness. On the other hand, this drawback can be at least partially overcome by an appropriate drug formulation [6]. Varona et al. demonstrated that the activity of lavandin oil could be enhanced by encapsulation due to the protection and controlled release of the oil components [27]. According to the literature, antimicrobial activity is usually correlated with phenolic, aromatic, or alcoholic components of essential oils. Their main mechanism of action is related to some disruption of the cell membrane and its increased permeability [32,33,34,35]. The main oil component of *L. x intermedia*, linalool, can destroy bacterial cell walls, change membrane potential, and enhance membrane leakage [36]. The antibacterial effect of EOs is generally more pronounced in the case of G+ bacteria. It is believed that the rigid outer membrane of G– bacteria limits the diffusion of hydrophobic compounds, therefore, protecting them against the harmfulness of EOs components [10]. However, the presented review of the studies of lavandin oil does not always confirm this general belief. Diverse antibacterial powers were observed regardless of gram-positive or negative attribution of studied bacteria.

#### 2.1.2. Other Biocidal

Articles reporting on the biocidal power of lavandin preparations, other than the antimicrobial power, are not numerous. They are compiled in Table 2. An antiparasitic activity of lavandin was indicated by Moon and coworkers. The scientists studied the effect of lavandin and true lavender essential oils on three protozoal pathogens: *Giardia duodenalis*, *Trichomonas vaginalis*, and *Hexamita inflata*. They demonstrated that oil concentrations of 1% or less could eliminate pathogens in the culture. *L. angustifolia* essential oil presented a slightly stronger effect than *L. x intermedia*. The authors stated: “Whether lavender essential oils can be used as a viable treatment for infected water sources or as a treatment of mammalian and/or fish parasitic infection is unknown. The previously unreported finding that these oils are potent anti-protozoan agents should, however, be further investigated” [37]. According to our best knowledge, there are no further reports on the antiprotozoal activity of *L. x intermedia* EO.


molecules-28-02986-t002_Table 2Table 2Other than the antimicrobial biocidal activities of lavandin natural products reported in the literature. Only original research articles were considered.Targeted OrganismsStudied AgentActivityOutcomeCitationHuman protozoal pathogens: *Giardia duodenalis*, *Trichomonas vaginalis*; Fish pathogen *Hexamita inflata**L. x intermedia* ‘Miss Donnington’ and *L. angustifolia* EOsantiprotozoalLI and LA EOs reduced the viability of all three parasites by causing their cell lysis. Oil concentrations of 1% or less could eliminate in vitro *T. vaginalis*, *G. duodenalis*, and *H. inflata*. At lower levels (0.1%), *L. angustifolia* was narrowly more effective than *L. x intermedia* against *G. duodenalis* and *H. inflata*. Microscopic analysis suggested that the mode of action of a substance may be via cell lysis, but further studies were needed to confirm the exact mechanism.[37]Root-knot nematode *Meloidogyne incognita*, walnut root lesion nematode *Pratylenchus vulnus**L. x intermedia* EO ‘Abrial’, ‘Rinaldi Ceroni’, ‘Sumiens’nematicidalStrong toxicity of all tested EOs to nematodes was indicated in vitro, even at low concentrations. The oil of ‘Rinaldi Ceroni’, with LC_50_ equal to 1.2 and 3.1 μg/mL for *M. incognita* and *P. vulnus*, respectively, was more potent than that of ‘Abrial’ and ‘Sumiens’, with 24.9 and 7.4, 17.4 and 0.5 μg/mL, respectively. When compared to the chemical positive control oxamyl, the EOs showed much higher toxicity only after a 4-h exposure. However, with incubation time, this difference was diminished and not evident after 24-h exposures. Activity differences among the cultivars were less apparent in vivo studies in soil. All studied lavandin EOs also significantly decreased *M*. *incognita* eggs density and their hatchability, as well as significantly reduced gall formation on tomato roots. Overall, oil treatments with the lavandin EOs positively impacted tomato plant growth.[38]Root-knot nematode *M. javanica**L. x intermedia* ‘Super’ hydrolate and other plants’ hydrolatesnematicidalHydrolate from *L. x intermedia* induced 100% mortality of *M. javanica* after a 24-h incubation while hydrolates from *Thymus vulgaris* and T. *zygis* were ineffective in this specific test. The study showed potent nematicidal effects of LI hydrolate against *M. javanica* (second-stage juvenile mortality and suppression of egg hatching) in vitro with LC_50_ at 16.9% and LC_90_ at 25.6%. It was found that the nematicidal active components were present in the aqueous fraction of hydrolate, which is in contrast to *Thymus* spp., where the organic fraction was more active.Tomato seedling tests in vivo, at sublethal doses, induced a significant reduction of nematode infectivity, significantly suppressing their penetration and host-plant root colonization with respect to their untreated controls. The tomato plant growth was not affected by these treatments.[39]Pine Wood Nematode *Bursaphelenchus xylophilus**L. x intermedia* EO of French origin and other EOsno nematicidalNo nematicidal action of LI essential oil was reported. In the study including 26 different EOs, only three of them: ajowan (*Trachyspermum ammi*), allspice (*Pimenta dioica*), and litsea (*Litsea cubeba*), showed nematicidal activity with LC_50_ ranging from 0.4 to 0.63 mg/mL.[40]Insects: *Leptinotarsa decemlineata*, *Spodoptera littoralis*, *Myzus persicae*, *Rhopalosiphum padi*; nematode *Meloidogyne javanica*; tick *Hyalomma lusitanicum*; plants: lettuce *Lactuca sativa*, English ryegrass*Lolium perenne**L. x intermedia* EO ‘Super’no nematicidalinsecticidalixodicidalallelopathicThe study showed the lack of nematicidal activity of *L. x intermedia* EO against *M. javanica*.It had an antifeedant effect against *S. littoralis* but not against other tested insects. The biocidal impact of this EO was strong in the case of *S. littoralis* (LC_50_ 25 μg/cm^2^) and *H. lusitanicum* (LC_50_ 28 μg/mg cellulose).The LI ‘Super’ oil did not show adverse effects on the germination of *L. sativa* and *L. perenne* seeds but significantly affected the growth of the root of *L. sativa*.[41]Confused flour beetle *Tribolium confusum* (*Coleoptera*); Radish plant *Raphanus sativus**L. x intermedia* hydrolate from stems and flowersinsect repellentallelopathicBehavioral tests with *T. confusum* showed good repellencies for both flowers and stems LI hydrosols. They exhibited (70–90% repellence for a dose of 12 μL/cm^2^ and RD_50_ = 3.58 and 3.26 μL/cm^2^ for flowers and stems, respectively). However, its repellent effects were lower than the one of the synthetic repellent MR-08 (RD_50_ = 0.001 μL/cm^2^).Regarding the allelopathic activity—both hydrolates inhibited the seed germination of *R. sativus*. Inhibition was higher in the case of the flower hydrolate (percentage of germination GP = 0%) than stem hydrolate (GP = 24%).[42]Confused flour beetle *Tribolium confusum**L. x intermedia* and other plants’ EOsinsecticidalLavandin oil and other tested oils, except *Origanum vulgare*, exhibited strong toxicity to larvae (LD_50_ 1.8, 59.7 or 109.9 μL/L air), pupae (37.3 and 38.7), and adults of *T. confusum* (29.8–81). The fumigant toxicity of these essential oils depended on the insect’s developmental stage.[43]Insects: *Leptinotarsa decemlineata*, *Spodoptera littoralis*, *Myzus persicae*, *Rhopalosiphum padi*; fungi: *Fusarium moniliforme*, *F. oxysporum*, *F. solani*; plants: *Lactuca sativa* and *Lolium perenne**L. x intermedia*, *L. angustifolia*, and other plants’ EOsinsecticidalallelopathicantifungalThe essential oils studied had antifeedant effects against the majority of targets tested. The most active oils were of *L. latifolia* and *T. vulgaris*. LI essential oil showed, at a dose of 100 μg/cm^2^, 86% and 88% feeding inhibition for *L. decemlineata* and *S. littoralis*, respectively, as well as 96% and 85% settling inhibition in the case of *M. persicae* and *R. padi*.Some phytotoxic activity against *L. sativa* and *L. perenne* was observed. The oils of *Lavandula* spp. were the most active against *L. sativa*, while *T. vulgaris* oil was the most phytotoxic against *L. perenne*. In both species, the germination percentage was more affected than the root and leaf length.Antifungal action against studied *Fusarium* spp. was indicated (EC_50_ 0.3–1.1 mg/mL).[44]Spotted wing *Drosophila suzukii**L. x intermedia* EOs ‘Grosso’ and ‘Provence’, *L. latifolia* and other plants’ EOsinsecticidaloviposition deterrentLI oil exhibited an insecticidal effect with EC_50_ ranging from 5.22 to 9.01 μL/L air in fumigation toxicity assays and 0.86–12.58% in contact toxicity assays. Linalool was the most effective monoterpene in fumigation assays, and *L. latifolia* essential oil was found to be the most effective whole essential oil (EC_50_ 3.28–4.21 μL/L air). In contact toxicity assays, 1,8-cineole was the most effective monoterpene, and *L. latifolia* EO was the most effective of all tested oils with EC_50_ of 0.69%.[45]Insects: *Sitophilus zeamais*, *Cryptolestes ferrugineus*, *Tenebrio molitor**L. x intermedia* (Italian origin) and other plants’ EOsinsect repellentLavandin oil showed a repellent activity for *S. zeamais and C. ferrugineus*, but the effect on *T. molitor* was less evident.[46]Bean weevil *Acanthoscelides obtectus**L. x intermedia* and other plants’ EOsinsecticidalAll tested essential oils exhibited robust activity against *A. obtectus* adults, with varying LC_50_ ranging values from 0.5–19 mg/L air depending on insect sex and the type of the essential oils. Lavandin and rosemary leaf essential oils were the most active (LC_50_ 0.5–2.4 mg/L air), followed by EO from lavender flowers (LC_50_ 1.9–3.7 mg/L air). Eucalyptus EOs had weaker fumigant activity with LC_50_ of 3–19 mg/L air.A positive correlation between total oxygenated monoterpenoid content and insecticidal activity was observed. Terpinen-4-ol, camphor, and 1,8-cineole alone exhibited the lowest LC_50_ values.[47]Plants: annual ryegrass *Lolium rigidum*, canola *Brassica napus*, wheat *Triticum aestivum*, subterranean clover *Trifolium subterraneum*Aqueous extracts of *L. x intermedia* ‘Grosso’ and other lavender speciesallelopathic*L. x intermedia* was the most phytotoxic among tested *Lavandula* species. It was determined that the stem and leaf extract of LI ‘Grosso’ significantly reduced the root growth of several tested plant species. The growth of *L. rigidum* roots was completely inhibited with an extract concentration of 10%. The fraction consisting of coumarin and 7-methoxycoumarin was the most phytotoxic. Coumarin was supposed to be the most phytotoxic and responsible for the observed phytotoxicity of the lavandin extract. Soil trials were conducted using the coumarin standard and the lavandin extract. In both cases, shoot lengths and weights were significantly reduced by a post-emergence application at all concentrations tested.[48]


Lavandin oil was also investigated regarding nematicidal power. D’Addabbo et al. observed a powerful biocidal effect on pathogenic root-knot nematode—*Meloidogyne incognita* and walnut root lesion nematode—*Pratylenchus vulnus*. The activity was so high (LC_50_ equal to 1.2 and 3.1 μg/mL for one essential oil of one LI cultivar) that authors postulated the oil as a component of the new nematicidal formulations alternative to synthetic nematicides. The effectiveness of lavandin oils of three different cultivars was evaluated both in vitro and in vivo in soil. LI oils significantly reduced parasite eggs’ density, their hatchability, and the gall formation on roots, overall positively impacting the growth of the studied tomato plants [38]. A potent nematicidal effect was also documented in vitro and in vivo by Andrés et al. [39]. They studied the lavandin and some other plants’ hydrolates, by-products produced during steam distillation, against the root-knot nematode *Meloidogyne javanica*. All tested hydrolates showed nematicidal effects in vivo (both on larvae mortality and suppression of egg hatching). The nematicidal active components of lavandin hydrolates were found to be present in the aqueous fraction, indicating that some polar constituents of lavandin, rather than those unpolar, are responsible for the observed effect. This is supported by the lack of nematicidal activity of *L. x intermedia* essential oil against the same nematode—*M. javanica* observed by de Elguea-Culebras et al. [41]. No nematicidal effect of lavandin oil was also observed by Park and coworkers against pinewood nematode (*Bursaphelenchus xylophilus*), but they did not present the detailed results for lavandin it and focused only on three chosen essential oils from other plants [40]. Considering all the above, the effectiveness of the nematicidal efficacy of lavandin is uncertain, and more studies are needed in this field.

The above-described nematodes are loss-making pests in agriculture. The other, an even bigger group of pests, is insects. Therefore, lavandin oils were also tested regarding their insecticidal or repelling properties for potential use as natural-based plant protection agents. The repelling properties of LI essential oils were indicated for the following insects: maize weevil *Sitophilus zeamais*, rusty grain beetle *Cryptolestes ferrugineus*, and yellow mealworm beetle *Tenebrio molitor* [46]. LI hydrosols also exhibited repellency in studies on the confused flour beetle *Tribolium confusum* [42]. This insect was also studied by Theou and coworkers [43]. They tested lavandin and other essential oils and found out that all tested oils, except oregano oil, exhibited strong toxicity to all developmental stages of the pest. They postulated the EOs as fumigants used for the protection of stored products in storehouses. The insecticidal power of lavandin oils was also recognized by other researchers who showed its efficacy against bean weevil *Acanthoscelides obtectus*, colorado potato beetle *Leptinotarsa decemlineata*, and spotted wing drosophila *Drosophila suzukii*. A positive correlation between total oxygenated monoterpenoid content and insecticidal activity was observed, with linalool and 1,8-cineole being the most effective terpenes [41,45,47].

Essential oils are also generally known for their allelopathic activity. In the case of lavandin oil, according to our knowledge, there are two studies concerning it. Both studies examined the effect of lavandin oil on lettuce *Lactuca sativa* and English ryegrass *Lolium perenne*. de Elguea-Culebras et al. showed low to moderate toxicity for the assayed oils in the allelopathic test. The LI essential oil did not show negative effects on the germination of *L. sativa* but reduced the growth of its root [41]. Santana and coworkers observed some phytotoxic activity against *L. sativa* and *L. perenne* seeds and observed negative effects on germination and growth [44]. Regarding hydrolates of lavandin, both hydrolates from flowers and stems were able to inhibit the germination of radish *Raphanus sativus*, with a stronger effect of the flower hydrolate [42]. Extensive allelopathic studies were conducted by Haig and coworkers, who studied the effects of aqueous extracts of several lavender species, including *L. x intermedia* [48]. The researchers indicated that *L. x intermedia* was the most phytotoxic among the tested species. It showed the effect on all four tested plant species. After fractionization of the LI extract, they found that the fraction consisting of coumarin and 7-methoxycoumarin was the most phytotoxic, and the coumarin was largely responsible for the effect. Coumarin is a well-known phytotoxin, and lavandin is known to contain more coumarins than, for example, *L. angustifolia* [48,49]. This explains the stronger effect of lavandin extracts as a phytotoxic agent.

### 2.2. Antioxidant Power

Oxidative stress is caused by an imbalance between the production and deactivation of oxygen-reactive species (ROS). ROS are naturally generated as by-products of oxygen metabolism and can play some physiological roles (among others—cell signaling). In stressful environmental conditions and the presence of xenobiotics, the production of ROS increases, leading to the imbalance that causes cell death and some pathologies, such as some cancers or neurodegenerative disorders. Therefore, apart from the endogenous antioxidant systems, also exogenous antioxidants are intensively studied. Antioxidants are compounds capable of slowing or retarding the oxidation of an oxidizable material and protecting organisms from oxidative stress. As synthetic antioxidants such as butylated hydroxyanisole (BHA) or butylhydroxytoluene (BHT) are suspected to be potentially harmful to human health, many natural products, including essential oil and plant extracts, have been investigated for their antioxidant properties [50,51,52].

Antioxidant power is the second, just after biocidal, activity reported in the literature for *L. x intermedia*. We have found ten original research articles reporting it: three of them relate to essential oil, six to plant solvent extracts, and one to both of the formulations. Most of them relate to in vitro studies based on simple radical scavenging assays. However, there is a report by Hancianu et al., who conducted a detailed study on Wistar rats subjected to scopolamine—an induced rat model of dementia [53]. The week-long inhalation of the essential oils of *L. angustifolia* and *L. x intermedia* by rats for a week induced some significant biochemical changes in their brains. Temporal lobe homogenates indicated increased activity of catalase (CAT), superoxide dismutase (SOD), and glutathione peroxidase (GPX), increased content of reduced glutathione (GSH), and reduced malondialdehyde (MDA) level. Rats in both lavender groups exhibited a significant decrease in MDA levels. MDA is one of the final products of polyunsaturated fatty acids peroxidation in the cells. Decreased MDA levels reflected the reduced lipid peroxidation caused by free radicals. Increased activity of CAT, SOD, and GPX, which create an enzymatic endogenous antioxidant defensive system [50], results in less oxidative stress. The results of the experiment show that lavender oils can be an indirect antioxidant. Indirect antioxidants enhance natural antioxidant defenses in living organisms by inducing the expression or increasing the activity of antioxidant enzymes [51]. Additionally, the authors reported that DNA cleavage patterns (present in the scopolamine-alone-treated rat group) were absent in the scopolamine-treated rats exposed to lavender oils, suggesting that lavender oils possess antiapoptotic and neuroprotective activity [53].

The antioxidant activity of lavandin EOs in vitro was indicated by Carrasco and coworkers [54,55]. In one paper, they described the studies of lavandin ‘Abrial’, ‘Super’, and ‘Grosso’ essential oils of Spanish origin. They conducted five different antioxidant assays: ABTS, DPPH, oxygen radical absorbance capacity (ORAC), chelating power (ChP), and reducing power (RdP). Studied *L. x intermedia* oils showed moderate antioxidant activities with varying activity for different cultivars and different tests. The authors also indicated mild inhibition of lipoxygenase (LOX) in vitro. LOX is a crucial enzyme in the transformation of arachidonic acid into leukotrienes which are involved in the occurrence of inflammation. LOX inhibition can lower leukotriene levels, thereby delivering an anti-inflammatory effect [54,56]. In the other article, Carrasco et al. studied the antioxidant activity and hyaluronidase inhibition of different species of *Lavandula* and *Thymus* essential oils. Hyaluronidase inhibitors can potentially serve as anti-inflammatory, anti-microbial agents, and anti-aging agents [57]. However, the researchers did not observe any inhibitory effect of lavandin oil and a very weak effect of LA oil on the enzyme. Regarding antioxidant activity, authors indicated some antioxidant activity of LI oils, in most assays, weaker than for true lavender oil, while both lavender oils showed significantly lower activity than *Thymus zygis*, which was caused by the high presence of thymol, the compound with known and strong antioxidant properties [55]. In general, phenolic agents act as antioxidants due to their high reactivity with peroxyl radicals, which are disabled by hydrogen atom transfer [51]. Lavender essential oils do not contain any significant amounts of phenolic terpenes and phenylpropanoids. In general, essential oils containing no or little phenols and cyclohexadiene-like components (e.g., γ-terpinene, α-terpinene, and α-phellandrene) do not exhibit significant direct antioxidant potency [51]. Direct antioxidants are compounds able to impair the radical chain reaction causing oxidation. However, despite the low presence of high-impact direct antioxidants, lavandin oil might show an indirect antioxidant property, as indicated by the above-described experiment of Hancianu et al. [53].

One more issue needs to be raised whenever the antioxidant activities of essential oils are assessed based on indirect methods such as the DPPH, ABTS, FRAP, or Folin–Ciocalteu tests. These methods are flawed models of antioxidant properties and are often inappropriate for reliable measuring antioxidant properties of essential oils. For example, a very popular and basic DPPH assay gives positive outcomes for some essential oils, not due to its antioxidant activity, but rather due to binding a hydrogen atom from C–H bonds from terpenes with a sufficiently low bond dissociation enthalpy such as α- or β-pinene and limonene. Therefore, the discoloration of the reactant might not necessarily indicate the antioxidant potency but also the presence of highly oxidizable compounds in the essential oil [51]. Moreover, the results are often difficult to compare due to multiple assays and different and inconsistent units. Thus, the discussion and most of the comparations are usually performed between samples from one experimental setup.

Lavandin ethanolic extracts exhibit higher antioxidant activity than lavandin oil [25]. Table 3, among other biological activities, presents a reported antioxidant activity for different kinds of lavender extracts. The superior to EO antioxidant power of the lavandin extract is caused by a different chemical composition compared, namely more abundant flavonoids, coumarins, phenolic acids, and their glycosides [25,49,52,58]. Higher activity was shown for extracts of post-distillation waste than the raw plant material before distillation [25,59]. Regarding the plant part, the most potent were found to be ethanolic extracts of leaves, then flowers and inflorescence stalk [60]. In one experiment, Berrington and Lall studied the antioxidant power of acetone LI and LA extracts, as well as other plant species. They have found that both lavender species had the lowest antioxidant activity (DPPH assay) among tested plant extracts (*Origanum vulgare*, *Rosmarinus officinalis*, *Thymus vulgaris*, *Petroselinum crispum*, *Foeniculum vulgare*, and *Capsicum annuum*) [61]. Whenever both activities of LI and LA were assessed in one study, they were rather similar, with varying antioxidant power depending on the assay and tested cultivar. True lavender extracts contained more phenolic acids and flavonoids, while coumarins were at higher levels in lavandin extracts [49,60,61]. Looking at the gathered studies, we cannot claim any superiority of *L. angustifolia* over *L. x intermedia* in this field.

### 2.3. Other Activities

Lavender essential oils are very popular in complementary/alternative medicine and are commonly used in aromatherapy to reduce stress, increase relaxation, and improve the quality of sleep [7,52,69,70]. The anxiolytic action of *L. angustifolia* oil was proven both in rodents and humans by both inhalation and ingestion. The most substantial evidence comes from the studies of Silexan, a patented active substance comprised of *Lavandula angustifolia* essential oil produced from flowers, standardized, compliant with European Pharmacopeia, and manufactured by Dr. Willmar Schwabe GmbH & Co. KG in Germany. Oral administration of Silexan impacted positively depressed mood, sleep disturbances, and the overall quality of life of clinical trial participants. It was also demonstrated that sleep improvement is a result of the anxiolytic effect, not the sedative effect per se [71,72,73]. The mechanisms underlying the anxiolytic effects of Silexan are not certain. Action through the mediation of gamma-aminobutyric acid (GABA) was proposed by Aoshima and Hamamoto [74]. Schuwald et al. demonstrated that Silexan inhibited voltage-dependent calcium channels (VOCCs) in neuronal cells at nanomolar concentrations [75]. It has been speculated that under anxiety or stress disorders, an enhanced calcium ions influx through VOCCs could increase the release of neurotransmitters such as glutamate and norepinephrine, which are involved in the pathogenesis of these diseases. Baldinger and coworkers showed that Silexan reduced the 5-HT1A receptor binding potential in the brain clusters such as the temporal gyrus, the fusiform gyrus, the hippocampus, the insula, and the anterior cingulate cortex. This led to an increase in extracellular serotonin content. Most probably, the effect of Silexan, apart from VOCCs inhibition, is additionally mediated via the serotonergic neurotransmitter system, particularly the 5-HT1A receptor, not through a GABAnergic mechanism [71,72,73]. The results of Siloxane might suggest a similar action of lavandin oil. However, there are almost few studies to support it. Hancianu et al. studied the effect of Romanian essential oils from *L. x intermedia* and *L. angustifolia*, as well as Siloxan, on a dementia rat model. They found that not only Siloxan but also both studied lavender oils acted neuroprotective and improved spacial memory and performance in various tests suggesting anxiolytic and antidepressant activity [53,65]. Regarding the influence on sleep, one human study with a mixture of lavandin oil with bergamot and ylang-ylang oils was described. The five-day-long aromatherapy improved the subjective assessment of sleep quality in the examined patients in cardiac rehabilitation [62].

Another often-raised biological action of lavender oils is their anti-inflammatory effect. Linalool, the main component of lavender oil, has been reported to have anti-inflammatory effects [7,52,76,77]. Huo and coworkers tested the action of this terpene on lipopolysaccharide(LPS)-induced production of inflammatory mediators such as tumor necrosis factor α (TNF-α) a and interleukin 6 (IL-6) [78]. LPS is a component of the outer membrane of G– bacteria, which triggers a strong immune response. Scientists have found that linalool reduced the production of TNF-α and IL-6 both in stimulated macrophages in vitro and in vivo in lung injury mouse models. They showed that linalool treatment attenuated lung histopathology in mice. In search of molecular mechanisms of the linalool action, the researchers investigated the phosphorylation of some proteins in NF-κB and MAPK pathways. Nuclear factor-kB (NF-κB) is the critical dimer protein controlling the expression of over 500 genes, including many inflammation-associated factors. Many agents and stimuli can activate NF-κB through canonical and noncanonical pathways. Usually, NF-κB is kept in the cytoplasm in inactive form due to its binding with its inhibitor (lκB). After some stimuli, such as LPS, IκB is phosphorylated and later degraded. The released NF-κB is translocated to the nucleus, followed by the p65 subunit phosphorylation, acetylation, methylation, as well as subsequent DNA binding and gene transcription. In this way, nuclear factor-kB activation mediates the activation of proinflammatory genes, including TNF-a and IL-6 [78,79]. Huo and coworkers indicated that linalool blocked LPS-induced IκBα phosphorylation and consequently prevented NF-kB activation. They also noticed reduced phosphorylation of ERK, JNK, and p38 in the MAPK signaling pathway, another effect that led to the anti-inflammatory activity of linalool. The effect of linalool on acute lung inflammation induced by other stress stimuli—cigarette smoke (CS)—was investigated by Ma et al. in vivo in mice [80]. It was indicated that linalool significantly reduced the production of TNF-α, and IL-6, along with some other inflammatory mediators. Overall, it inhibited the infiltration of inflammatory cells and lung inflammation. The researchers noted that the terpene suppressed CS-induced NF-κB activation by inhibiting CS-induced IκBα and p65 NF-κB protein phosphorylation. Therefore, the demonstrated effect of linalool is in agreement with the former research conducted by Huo et al. [78]. This mechanism might be responsible for the reported anti-inflammatory properties of different *Lavandula* essential oils. Specifically to *L. x intermedia*, we have found a study on this activity—the above-mentioned study of Baker and coworkers on the lavandin oil effect on mice acute colitis induced by *Citrobacter rodentium* bacteria [14]. The oil administration lowered the expression of inducible nitric oxide synthase, interferon-gamma, interleukin 22, and macrophage inflammatory protein-2α gene expression and decreased neutrophil and macrophage infiltration. In colitic mice, oral gavage with lavandin oil resulted in milder disease, decreased morbidity and mortality, and reduced intestinal tissue damage. Barocelli et al. also noticed the gastroprotective effect of lavandin oil, as well as linalool and linalyl acetate, delivered separately. LI oil administration protected against acute ethanol-induced gastric ulcers in rats, but the mechanism of its protective action was not elucidated [64]. They have also investigated the analgesic effect of chemical and thermic stimuli. Lavandin oil, especially when inhaled, prolonged the response to unpleasant stimuli, suggesting an antinociceptive effect. However, some authors postulated that, instead of having a direct analgesic effect, inhalation of lavender oil may cause a more positive attitude and therefore alter the subjective perception of pain unpleasantness [7,81]. Linalool, the main lavender terpene, was demonstrated to induce analgesic effects in mice—significantly increasing the pain threshold and attenuating pain behaviors. Antinociceptive effects were absent in olfactory-deprived mice in which the olfactory epithelium was damaged. Thus, the action of linalool might be triggered by the olfactory sensory input. An immunohistochemical study revealed that linalool activated hypothalamic orexin neurons, crucial mediators for pain processing. Still, the actual mechanism is not understood [82,83].

Table 3 also presents the work by Ballabeni and coworkers, who also indicated the antithrombotic and anti-platelet activity of lavandin oil both in vitro and in vivo. The oil administration significantly reduced thrombotic events in mice models of pulmonary thromboembolism variance with aspirin used as a reference drug but without inducing hemorrhagic complications as acetylsalicylic acid. Regarding the potential anticancer properties of lavandin, there are only a few investigations. Berrington et al. studied the acetone extracts of LI and LL on the cervical epithelial carcinoma cell line and observed no anticancer activity [61]. Tabatabaei et al. observed the potential anticancer/antiproliferative activity of lavandin oil on the human breast cancer cell line MCF-7 [67], and Ovidi and colleagues found similar activity on different cancerous cell lines [66]. The latter authors also found that nanoformulation of the essential oil increased its antiproliferative activity.

As already mentioned, the biological effects of lavandin oil, which have been tested and verified, are mostly related to biocidal or antioxidant activity in vitro. These effects do not represent the full potential of its action In vivo. Lavandin oil is widely used in aromatherapy and massages. Despite this, the therapeutic effects of this oil have been largely overlooked in scientific studies when compared to its parent species. When *L. x intermedia* was studied and compared to *L. angustifolia* in terms of the biological/therapeutic effects, it usually gave a similar performance, in the case of antimicrobial—even more powerful. The main components of both lavender oil—linalool and linalyl acetate, which is quickly metabolized in vivo to linalool [84]—affect molecular pathways and induce some biological activities. Thus, we can expect similar biological activities in general. The differences in composition lie in the secondary and trace constituents, such as camphor, 1,8-cineole, and borneol, which sum up to approximately 30% of the oil. These terpenoids are usually alleged of increased the biocidal action of LI and LL, but increased biocidal properties are a double-edged sword. Of the three mentioned, especially camphor, despite its wide use in pharmacy, is considered to be potentially harmful to health, and the toxicity of camphor is well-documented [85]. Although most cases of camphor poisoning were due to oral ingestion, a few reports indicate that toxic doses of camphor can be absorbed through inhalation and skin contact too. It has been estimated that severe toxicity, which can cause convulsions, may occur in adults at a dose of around 34 milligrams per kilogram of body weight [86]. The typical signs of camphor poisoning when consumed by humans include headaches, nausea, vomiting, dizziness, muscle stimulation causing tremors and twitching, seizures, and delirium. The severity of these symptoms varies depending on the amount of camphor ingested [85]. Therefore, the United States Food and Drug Administration in 1983 set a limit of 11% in consumer products, Ph. Eur. allows the maximum dosage of *Camphora racemica* or *D-Camphora* as 10% when admitted topically [85,87]. It is possible that one of the reasons, but nowhere in the literature explicitly stated, why LI and LL are less popular in traditional medicine than lavender is the toxicity of camphor. On the other hand, camphor is appreciated in medicine and commonly used in topical drug formulations. Additionally, lavandin poisoning is uncommon. To the best of our knowledge, there are no described cases of lavandin poisoning in the literature, with the exception of one case involving an 18-month-old infant who ingested a homemade lavandin extract [88]. Unfortunately, the formulation and method of preparation of this extract were not described. However, the authors detected linalyl esters and acetone in both the extract and the patient’s blood, suggesting that it may have been an acetone extract.

Secondary and minor components can influence the overall biological action of linalool. They may be indifferent or interact with each other leading to synergistic or antagonistic effects. Therefore, the activity of essential oil can sometimes be stronger than that of it its main components. It is generally believed that minor chemicals play a critical role in synergistic activity. The compositional complexity and natural variability of the plant material make this kind of research challenging.

## 3. The Use of Lavandin in Industry and Everyday Life

Despite the numerous biological effects of lavandin, which have been documented and described in the last chapter, the majority of lavandin usage, both in industry and everyday practice, is just due to its aroma. Albeit the lavandin is also cultivated for its decorative values and for obtaining herb/dried plant material, it is widely cultivated mainly for its essential oil, and this raw material prevails in use. Indeed, lavandin oil found its use in the perfume industry, the soap industry, cosmetics, and aromatherapy [54,89,90,91,92,93,94,95,96,97]. It is also used as a fragrance in a variety of household products (detergents, room sprays, and industrial perfumes) and food beverages because of its fresh and herbaceous odor and availability [89,90,96,98,99,100].

### 3.1. Fragrance-Related Usages

As mentioned in the first part, there are many cultivars of lavandin in cultivation for its essential oil, among others ‘Abrial’, ‘Grosso’, ‘Super’, ‘Hidcote Giant’, ‘Dutch’, ‘Grappenhall’, ‘Provence’, ‘Seal’, ‘Sumian’, and ‘Budrovka’ [90,94,101,102]. The ‘Abrial’ is highly valued for its fragrance—similar to that of true lavender, while ‘Grosso’ is currently the most cultivated for economic reasons. Nevertheless, the ‘Grosso’ scent may not be at first as pleasant as that of the ‘Abrial’, but the heaviness and truculence of the ‘Grosso’ may be an asset—depending on the purpose of its use [93]. The lavandin oil is a pale yellow to almost colorless liquid and has a strong herbaceous odor with a fresh camphene-cineole-like top note. Interestingly, in the perfume or fragrance industry, under the heading of “lavender notes”, the whole group is hidden: lavender oil itself, spike lavender oil, and lavandin oil. A small percentage of “lavender” supports the complex of bergamot and oak moss within a great number of Chypre-type perfume compositions (e.g., 5% of lavandin oil). Higher percentages are found in Fougére-type perfumes (Table 4) which often belong to one of the most successful market segments [100]. In general, lavandin oil is used in large quantities for a fresh note in perfumes. It blends well with natural and synthetic perfumery products such as citronella, cypress, decyl alcohol, geranium oil, pine needle oil, oregano oil, patchouli, and thyme oil) [96]. In toilet soap fragrances, for instance, lavandin oil is used as the only fragrance [100]. It also goes well with detergent products [96].

Another important use of lavandin is the production of absolute. Absolutes are indispensable ingredients in perfumery. The lavandin absolute is a dark green viscous liquid of herbaceous odor (sweeter than the essential oil), reminding the flowering lavandin. It has been used for herbaceous, Fougére, new-mown-hay types, floral fragrances, forest notes, and refreshing colognes. It blends well with clove oil, bergamot, lime, and patchouli and softens rough ionones [96].

According to the International Nomenclature of Cosmetic Ingredients (INCI), lavandin essential oil should be indicated as Lavandula Hybrida Oil or Lavandula Intermedia Oil in the composition of cosmetics. However, in the official documents of the European Commission (cosmetic ingredients database: CosIng) [103], lavandin oil appears under several names: two already mentioned (Lavandula Hybrida Oil and Lavandula Intermedia Oil), which refer to the general name of the species, in addition to other names which refer to lavandin varieties: Lavandula Hybrida Abrial Herb Oil, Lavandula Hybrida Barreme Herb Oil, Lavandula Hybrida Grosso Herb Oil (Table 5).

According to the official User Guide [104], CosIng is “the online consultation tool of the European Commission describing cosmetic ingredients contained in:Cosmetics Regulation (EC) No 1223/2009 of the European Parliament and of the Council;the Inventory of Cosmetic Ingredients, as amended by Decision 2006/257/EC establishing a common nomenclature of ingredients employed for labeling cosmetic products throughout the EU; andopinions on cosmetic ingredients of the Scientific Committee on Consumer Safety.”

Table 5 was prepared based on the CosIng ingredient index. It contains a list of lavandin cosmetic raw materials with a short description. INCI names containing “Lavandula Hybrida” refer to oils, extracts, and hydrosols obtained from lavandin flowers. In turn, those with “Lavandula Intermedia” in the name refer to products obtained from the whole herb (leaves, flowers, stems). It is worth noting that the flowers themselves are also used as a cosmetic ingredient [103]. In the USA, cosmetic ingredients are laid down in the Title 21 of the Code of Federal Regulations (CFR), reserved for rules of the Food and Drug Administration. There, it is stated that essential oils, oleoresins (solvent-free), and natural extractives (including distillates) are generally recognized as safe, for their intended use, within the meaning of section 409 of the Act [105].

Is LI oil a popular cosmetic raw material in general, and is it often used in cosmetic formulations? It is difficult to estimate as there are no available registers that contain reliable data of such kind. We performed a rough evaluation based on Internet resources, mainly the Environmental Working Group (EWG) and INCI Decoder (science-based ingredient verifying tool). EWG (Environmental Working Group) is an independent nonprofit organization operating in the US that issues various product safety warnings and customer guides. It collected information about 504 different cosmetics containing lavandin oil on the US market, and the summary based on its register is presented in Table 6 [106]. It did not differentiate between different lavandin cosmetic raw materials. According to the information given by INCI Decoder, there were 257 cosmetics with Lavandula Hybrida Oil, 20 with Lavandula Hybrida Grosso Herb Oil, 8 with Lavandula Hybrida Herb Oil, 3 with Lavandula Hybrida Abrial Herb Oil, and 2 with Lavandula Hybrida Abrialis Oil [107]. When analyzing this data, it can be noticed that cleansing products predominate. In second place are diverse types of skincare products. Scented candles containing this oil are also available for sale [108,109]. Naturally, perfumes and Eau de Toilettes containing this ingredient should also be mentioned [110,111,112,113].

### 3.2. Medicinal/Therapeutic Use

As extensively discussed in Chapter 2, lavandin and lavandin oil have a number of scientifically proven biological properties. Although not listed in the European Pharmacopoeia as a traditional medicinal plant, it is recognized by WHO [87,114]. Currently, lavandin oil is widely used in aromatherapy. As an alternative to synthetic formulations, essential oil blends are used for natural disinfection and room freshening. Lavandin-based products with claims of antidepressant, anxiolytic, analgesic, neuro-, and gastroprotective effects, as well as improved memory and sleep, can be purchased. These aromatherapeutic claims can be found on many cosmetics labels, such as bath and massage oils. They are not entirely unfounded (as shown in the previous chapter). However, they should be treated cautiously, as lavandin oils have not been subjected to extensive clinical trials.

Medical applications of essential oils, e.g., treatment of carcinoma cells, are generally limited by the lack of their solubility in water. According to recent studies, this problem could be solved by introducing EOs as nano-formulations [115]. Lavandin EO, specially formulated as nanoemulsions, exhibits pronounced cytotoxic effects on human neuroblastoma cells, human lymphoblastic leukemia cells, and human colorectal adenocarcinoma cells [66].

It is worth emphasizing that despite the widespread use of lavandin, there is still not enough research to extend our knowledge about its medical properties, contrary to true lavender.

### 3.3. Other Uses

It is worth emphasizing that lavandin is not only a raw material to produce cosmetic ingredients. Lavandin oil is used as a natural flavorant in baked goods, frozen dairy, gelatin, soft candy, pudding, alcoholic beverages, and other food products [116]. *L. x intermedia* flowers are also a source of nectar from which bees produce honey. As might be expected, honeys derived from different lavender species show specific volatile compounds profile. Lavandin honey can be easily distinguished from the honey of other lavenders due to its high content of phenylacetaldehyde [117]. As a curiosity, lavandin is sold as a “medicinal plant for bees” or other pollinators such as bumblebees or butterflies and moths—it is considered a forage plant for pollinators [118,119,120].

Lavandin is also utilized in landscaping and agrotourism. Entrepreneurs started offering tourists opportunities such as visiting a lavender farm, where people can enjoy its beauty, learn about lavender cultivation, pick up bouquets, taste lavandin honey and ride horses in the area of lavender fields. Some farms also possess small-scale distillation units and present an opportunity to participate in the oil distillation process. Lavandin is even preferred here over true lavender due to its larger, decorative flowers that are very attractive for creating dry bouquets and are beautiful scenery for photographing and photo sessions [121,122].

*L. x intermedia* was also introduced in olive groves in Spain as a complementary crop that can help fight erosion, support biodiversity, and foster sustainable development. It was a result of the European Commission’s Horizon 2020 project, Diverfarming, led by scientists from the University of Córdoba. The project addressed food security, sustainable agriculture silviculture, bioeconomy, and water management [123].

Scientists are trying to find other uses for LI. Essential oils are highly soluble in supercritical carbon dioxide, and therefore the supercritical fluid can be an effective fluid medium for impregnating these compounds into a polymer. Varona et al. created a supercritical impregnated *n*-octenyl succinate-modified starch with lavandin oil. The product obtained in this way may be used as a substitute for synthetic drugs in livestock [124]. This solution is considered for the production of drugs with controlled release of active compounds. In a review, Lesage-Meessen et al. indicated that distilled straws of lavandin (the byproduct of oil extraction) may be up-cycled as valuable raw material in biotechnological processes to obtain products with antimicrobial, antioxidant activity sought by the pharmaceutical and cosmetic industry [125].

Products based on lavandin are very common in our lives. It is possible that when we wash our hands with “lavender” soap or use “lavender” body lotion, we are using, in fact, products based on lavandin material. It is worthwhile to think about the importance of lavandin and give more credit to the “bastard”, as it is unfavorably called.

## 4. Conclusions

In this review article, we comprehensively present and discuss the biological activities of essential oil and other active ingredients from *Lavandula x intermedia* (LI), as well as its current typical uses in industries and everyday life. Lavandin oil is primarily used in cosmetics. For therapeutic purposes, it is used in aromatherapy sessions, although, except for WHO, LI oil is not recognized officially as any medical agent. Thus, *Lavandula angustifolia* (LA) essential oil, a raw material acknowledged by Ph. Eur., receives the most attention from the scientific community and has been studied more extensively for its biological effect and possible use in therapy. LI and LA essential oils have a similar chemical composition, with some differences, mainly in camphor, borneol, and 1,8-cineole. This can potentially increase the risk of lavandin toxicity due to its higher camphor content and lead to some differences in oil biological activities, starting from the smell and ending with biocidal effects (similar or slightly stronger for lavandin). This paper summarizes all reported biological activities of LI, including antioxidant, biocidal, anxiolytic, neuroprotective, antithrombotic, immunomodulatory, and analgesic effects. In studies, when the *L. x intermedia* and *L. angustifolia* or even Silexan were compared, the effects were usually similar. This implies that *L. x intermedia* possesses similar therapeutic potential as its parent species. However, it cannot be definitively confirmed as the research is scarce. Currently, there is not enough evidence, and more studies are needed in this area. Accordingly, the answer to the question stated in the introduction, “Is lavandin oil less valuable than true lavender oil in other non-perfumery applications?” cannot be answered with 100% certainty at this time. *L. x intermedia* is as good as true lavender for its biocidal action, similar in terms of antioxidant properties, and possibly possesses other therapeutic values as *L. angustifolia*. With all certainty, lavandin is superior to *L. angustifolia* in terms of yields of the essential oil. This practical aspect made lavandin a dominant lavender plant cultivated all over the world.

## Figures and Tables

**Table 3 molecules-28-02986-t003:** Bioactivities of lavandin natural products reported in the literature (other than the biocidal). Only original research articles were considered.

Studied Agents	Methods of Assessment	Observations, Conclusions	Activity	Citation
A mixture of *L. x intermedia* ‘Super’, *Citrus bergamia* (bergamot), and *Cananga odorata* (ylang-ylang) EOs	Pittsburgh Sleep Quality Index (PSQI), a human-randomized, double-blind crossover study	The goal of the investigation was to determine if there was a significant difference between the sleep quality of patients in cardiac rehabilitation who inhaled a placebo and those who inhaled an aroma (for five consecutive nights). The sleep quality of participants receiving the studied oil mixture was significantly better than that of participants receiving the placebo oil.	Improving sleep quality	[62]
*L. x intermedia* ‘Grosso’ EO, French origin	In vivo mice; in vitro guinea pigs and rats; platelet aggregation studies, clot retraction assay, tail transection bleeding	Antiplatelet properties of lavender oil were exhibited on guinea-pig platelet-rich plasma towards platelet aggregation induced by arachidonic acid, U46619, collagen, and ADP. Lavandin oil (100 mg/kg/day for five days) significantly reduced thrombotic events in mice models of pulmonary thromboembolism without inducing hemorrhagic complications at variance with aspirin used as a reference drug. Major components of the oil were also studied, but none of them triggered such effects as the oil.	antithrombotic	[63]
*L. x intermedia* ‘Okanagan’ and wild-type EOs	In vivo mice studies; survival and body weight measurements, histopathological scoring, bacterial count, immunofluorescence, RT-PCR	Lavandin oil was beneficial in the case of acute colitis induced by *Citrobacter rodentium* in mice (reduced morbidity and mortality). The oil lowered the expression of iNOS, IFN-γ, IL-22, and MIP-2α mRNA and reduced the inflammation compared with infected control mice. LI oil prevented severe cecal mucosal damage during the infection.	anti-inflammatory	[14]
*L. x intermedia* ‘Grosso’ EO, French origin	In vivo rats and mice study; acetic acid writhing test, the activity cage, hot plate tests, Rainsford’s method	Orally administered or inhaled lavandin oil reduced the writhing response to acetic acid treatment. Inhalation of LI oil produced an inhibition of the hot-plate response proportional to the time of exposure to oil vapors. Moreover, when taken orally, lavandin oil, as well as its separate main components—linalool and linalyl acetate oral administration protected against acute ethanol-induced gastric ulcers.	analgesic gastroprotective	[64]
*L. x intermedia*, *L. angustifolia* EOs (Romanian origin) and Silexan	In vivo rats’ studies; behavioral tests: Y-maze task, elevated plus-maze task, forced swimming test, radial arm-maze task	The subjects were Wistar rats subjected to scopolamine—an induced rat model of dementia. Daily respiratory exposure for a week of all three tested lavender oils reduced anxiety and depression (based on elevated plus-maze and forced swimming tests). Moreover, the performance in Y-maze and radial arm-maze tests was improved, suggesting positive effects on spatial memory.	anxiolytic antidepressant improve spatial memory	[65]
*L. x intermedia*, *L. angustifolia* EOs (Romanian origin) and Silexan	In vivo rats’ studies; Superoxide dismutase (SOD), glutathione peroxidase (GPX), and catalase (CAT) specific activities, the total content of reduced glutathione (GSH), malondialdehyde (MDA) level (lipid peroxidation), and DNA fragmentation assays	Potent neuroprotective effects of both lavender oils against scopolamine-induced oxidative stress in the rat brain were shown. The subjects were Wistar rats subjected to scopolamine—an induced rat model of dementia. The EOs were administered an electronic vaporizer in a Plexiglas chamber. Daily subacute exposure for a week to the EOs significantly increased the activity of antioxidant enzymes (SOD, GPX, and CAT), increased the content of reduced GSH, and reduced lipid peroxidation (MDA level) in rat temporal lobe homogenates, suggesting antioxidant potential. DNA cleavage patterns were absent in the lavender groups, suggesting anti-apoptotic activity. The substantial antioxidant and antiapoptotic effect of EOs has been recognized as a cause of neuroprotective effects against scopolamine-induced oxidative stress in the rat brain.	antioxidant antiapoptotic neuroprotective	[53]
*L. x intermedia* ‘Grosso’ and ‘Super; *L. angustifolia*, *L. latifolia*, and other *Thymus zygis* and *Thymus hyemalis* EOs	DPPH, ORAC, chelating power test, nitric oxide scavenging capacity, reducing power, TBARS, hyaluronidase inhibitory activity	LI oils had the most potent chelating power among tested EOs. It was explained by the high contribution of ester and ether groups in their essential oils. The oils displayed significant values in hydroxyl and peroxyl radical scavenging assays, however lower than LA oil. That was explained by the high concentration of alcohol and ester groups, mostly linalool and linalyl acetate. Both IL essential oil also showed weaker performance in DPPH and ABTS tests than LA oil and substantially lower than the performance of *T. zygis* EO due to its high thymol content. No inhibition of hyaluronidase by lavandin essential oil was observed compared to weak for *L. angustifolia* and strong for *T. zygis* (high thymol chemotype).	antioxidant no hyaluronidase inhibition	[55]
*L. x intermedia* ‘Abrial’, ‘Super’ and ‘Grosso’ EOs, Spanish origin	ORAC, ABTS, DPPH, chelating power and reducing power; lipoxygenase inhibitory test	Studied *L. x intermedia* oils showed moderate antioxidant activities, especially due to the presence of linalool and linalyl acetate. A mild lipoxygenase inhibitory effect was observed, suggesting a potential anti-inflammatory effect of LI oils, mostly due to the presence of linalool and camphor.	antioxidant; anti-lipoxygenase (potential anti-inflammatory)	[54]
*L. x intermedia* ‘Grosso’ EO and its nanoformulations	MTT assay	Studied EO and its nanoformulations exhibited some antiproliferative activity on various tested cell lines. The EC50 values indicated that Caco-2 (human colorectal adenocarcinoma), MCF7 (human breast adenocarcinoma), and MCF10A (normal breast epithelial) cells were more resilient to the treatment, while CCRF-CEM (human lymphoblastic leukemia) and SHSY5Y (human neuroblastoma) cells were more responsive to it. Nanoformulation of lavandin oils demonstrated greater antiproliferative activity compared to EO, especially in cases of more resistant cell lines.	antiproliferative	[66]
*L. x intermedia* EO	MTT assay	Some cytotoxicity of *T. vulgaris*, *R. officinalis*, and *L. x intermedia* EOs on the MCF-7 (human breast cancer) cell line was observed in vitro.	anticancer	[67]
Acetone extracts of selected plant species, including *L. x intermedia* ‘Margaret Roberts’ and *Lavandula latifolia*.	DPPH, XTT (cytotoxicity test), microscopic tests on a noncancerous African green monkey kidney (Vero) cell line, and an adenocarcinoma cervical cancer (HeLa) cell line	Both studied lavender species had the lowers antioxidant activity among tested plant extracts (*Origanum vulgare*, *Rosmarinus officinalis*, *Thymus vulgaris*, *Petroselinum crispum*, *Foeniculum vulgare*, *and Capsicum annuum*) and showed minimal inhibition of DPPH. Tested lavender oils did not exhibit potent anticancer properties.	low antioxidant no anticancer	[61]
*L. x intermedia* ‘Super’ waste after distillation (methanol macerate), Spanish origin	TFC, HPLC-DAD, LC-ESI-MS/MS, DPPH, CL, XO	The lavandin waste material had a clear antioxidant activity, mostly due to phenolic content. Twenty-three phenolic compounds were identified by liquid chromatography in methanol macerates from lavandin waste, including phenolic acids, hydroxycinnamoylquinic acid derivatives, glucosides of hydroxycinnamic acids, and flavonoids. Lavandin waste was found to be a relatively poor source of rosmarinic acid, though it had high levels of chlorogenic acid and hydroxycinnamic acid glucosides, which are particularly active in scavenging hydroxyl radicals.	antioxidant	[58]
*L. x intermedia* ‘Bila’, ‘Budrovka SN’, ‘Budrovka’; EOs and ethanolic extracts of flowers before distillation and of post-distillation waste material; grown in Croatia	DPPH, LC-PDA-ESI-MS	Lavandin ethanolic extracts exhibited higher antioxidant activity than EOs. Higher activity was shown for extracts of post-distillation waste than flowers before distillation (more than 90% activity at a concentration of 1 mg/mL). To achieve a similar effect with flower extracts and EOs, a concentration of 2.5 or 40 times higher, respectively, had to be used. The better antioxidant activity of the ethanolic extract was attributed to their different chemical composition compared to EOs. The main components of both types of lavandin ethanol extract were hydroxycinnamic acid glycosides and essential oils—linalool, linalyl acetate, 1,8-cineole, and camphor. Extracts of post-distillation lavandin waste contained a relatively higher concentration of identified compounds than of fresh flowers, which could explain their better antioxidant activity. The tested cultivars ‘Bila’, ‘Budrovka SN’, and ‘Budrovka’ showed twice as much antioxidant activity as compared to the ‘Sumiens’, ‘Super A’, and ‘Grosso’ tested by other researchers.	antioxidant	[25]
*L. x intermedia* ‘Budrovka’, and *L. angustifolia* flower, inflorescence stalk, and leaf ethanolic extracts, Croatian origin	DPPH, iron chelating activity, reducing power, lipid peroxidation inhibition, total antioxidant capacity assays; HPTLC	Ethanolic extracts of leaves of LI and LA were mainly the most active in terms of antioxidant power among the studied plant parts. Flower extracts were slightly weaker, and inflorescence stalk extracts had the lowest antioxidant activity. The performed antioxidant tests, except total antioxidant capacity, demonstrated that lavandin ‘Budrovka’ extracts were slightly less potent than those of *L. angustifolia*. It was justified by their lower polyphenolic contents. Rosmarinic acid was the most abundant polyphenolic constituent of both tested *Lavandula* extracts and was considered the main contributor to the exhibited antioxidant power. The authors concluded that *L. x intermedia* ‘Budrovka’ is as potent an antioxidant as *L. angustifolia*.	antioxidant	[60]
Macerates, decoctions, and other extracts from *L. x intermedia* ‘Grosso’, ‘Gros Bleu’, and *L. angustifolia*, Polish origin	TPC, TPF, HPLC-DAD/MS, DPPH, FRAP	HPLC analysis showed the presence of phenolic acids (rosmarinic acid, ferulic acid glucoside, caffeic acid, ellagic acid), flavonoids (morin, isoquercitrin, vanillin), and coumarins (herniarin, coumarin). The content of phenolic acids and flavonoids was higher in lavender extracts, while coumarins were at higher levels in lavandin extracts of both cultivars, regardless of extract type. The highest radical scavenging and reducing properties were observed for ultrasonic-assisted extracts. Aqueous-ethanolic extracts (UAE and macerates) showed more antioxidant power than aqueous extracts (decoctions). No significant difference between the species was observed.	antioxidant	[49]
Different extracts of *L. x intermedia*, *Lavandula latifolia*, and other plant species	DPPH, NBT/hypoxanthine superoxide, CL, β-carotene bleaching test, TFC	Authors could not choose the best plant source of natural antioxidants among studied species because each plant species exhibited different antioxidant or scavenging activities. Regarding the plant material prior to distillation, *L. x intermedia* was most potent in most of the conducted tests. The ethyl acetate and dichloromethane fractions of extracts contained high phenolic content and radical scavenging activities.	antioxidant	[59]
*L. x intermedia* ‘Super A’, ‘Grey Hedge’, and *L. angustifolia* EOs	β-carotene bleaching test, DPPH, ABTS	The studied lavender oil samples showed high antioxidant activity. *L. x intermedia* ‘Super A’ showed the highest inhibition activity (IC50 89.81 ± 0.17 μg/mL) in the DPPH assay. Some cultivars of LA were the most potent in the remaining antioxidant activity tests.	antioxidant	[68]

**Table 4 molecules-28-02986-t004:** Examples of Fougére type fragrance compositions [100].

Formula 1	Ingredient	%
	Lemarone *	0.5
	Cumarin	1.0
	Fir resinoid absolute	1.0
	Orange Oil Florida	1.5
	Osmantinia Givco 140	1.5
	Geranium oil bourbon	1.5
	Ylang-ylang extra	2.0
	Oxyoctaline formate *	2.0
	Musk moskene	2.0
	Benzyl acetate	3.0
	Benzyl salicylate	4.0
	Dimetol *	4.0
	Amyl salicylate	5.0
	Isoraldeine 70 *	6.0
	Tetrahydrolinalool	8.0
	Linalyl acetate	10.0
	Bornyl acetate crystal	10.0
	Lavandin Oil (French)	10.0
	Alpha-Hexylcinnamic aldehyde	12.0
	Birch Leaf Givco 115	15.0
**Formula 2**	**Ingredient**	**%**
	Civet substitute 40/2	0.5
	Eugenol	3.0
	Sandalore *	3.0
	Benzoin resinoid Siam	4.0
	Vetiver oil Haiti	4.5
	Isoraldeine 70 *	7.0
	Linalyl acetate	7.0
	Tree moss resinoid 50	8.0
	Rhodinol 70 *	9.0
	Coumarin crystal	10.0
	Musk moskene	10.0
	Patchouli oil	14.0
	Lavandin Oil	20.0

* Registered trademark for Givaudan.

**Table 5 molecules-28-02986-t005:** List of official lavandin raw INCI names [103].

INCI Name	Description According to CosIng
Lavandula Hybrida Abrial Herb Extract	Lavandula Hybrida Abrial herb Extract is an extract obtained from the flowering herbs of the Lavandin, Lavandula hybrida var. abrial, Labiatae
Lavandula Hybrida Abrial Herb Oil	Lavandula Hybrida Abrial Herb Oil is an essential oil distilled from the flowering herbs of the Lavandin, Lavandula hybrida var. abrial, Labiatae
Lavandula Hybrida Barreme Herb Extract	Lavandula Hybrida Barreme Herb Extract is an extract obtained from the flowering herbs of the Lavandin, Lavandula hybrida var. barreme, Labiatae
Lavandula Hybrida Barreme Herb Oil	Lavandula Hybrida Barreme Herb Oil is an essential oil distilled from the flowering herbs of the Lavandin, Lavandula hybrida var. barreme, Labiatae
Lavandula Hybrida Extract	Lavandula Hybrida Extract is an extract of the flowers of Lavandin, Lavandula hybrida, Labiatae
Lavandula Hybrida Extract Acetylated	Lavandula Hybrida Extract Acetylated is an acetylated extract of the flowering herbs of the Lavandin, Lavandula hybrida, Labiatae
Lavandula Hybrida Flower	Lavandula Hybrida Flower is the flower of Lavandula hybrida, Lamiaceae.
Lavandula Hybrida Flower Extract	Lavandula Hybrida Flower Extract is the extract of the flowers of lavender, Lavandula hybrida, Labiatae
Lavandula Hybrida Flower Water	Lavandula Hybrida Flower Water is an aqueous solution of the steam distillate obtained from the flowers of lavender, Lavandula hybrida, Labiatae
Lavandula Hybrida Grosso Herb Extract	Lavandula Hybrida Grosso Herb Extract is an extract obtained from the flowering herbs of the Lavandin, Lavandula hybrida grosso, Labiatae
Lavandula Hybrida Grosso Herb Oil	Lavandula Hybrida Grosso Herb Oil is an essential oil distilled from the flowering herbs of the Lavandin, Lavandula hybrida grosso, Labiatae
Lavandula Hybrida Herb Extract	Lavandula Hybrida Herb Extract is an extract obtained from the flowering herbs of the Lavandin, Lavandula hybrida, Labiatae
Lavandula Hybrida Herb Oil	“Lavandin Oil”. Lavandula Hybrida Herb Oil is an essential oil distilled from the flowering herbs of the Lavandin, Lavandula hybrida, Labiatae
Lavandula Hybrida Oil	Lavandula Hybrida Oil is the essential oil obtained from the flowers of the Lavandin, Lavandula hybrida, Labiatae
Lavandula Intermedia Extract	Lavandula Intermedia Extract is an extract of the leaves, flowers, and stems of the Lavender, Lavandula intermedia, Labiatae
Lavandula Intermedia Flower/Leaf/Stem Extract	Lavandula Intermedia Flower/Leaf/Stem Extract is the extract of the leaves, flowers, and stems of Lavender, Lavandula intermedia, Labiatae
Lavandula Intermedia Flower/Leaf/Stem Oil	Lavandula Intermedia Flower/Leaf/Stem Oil is the volatile oil obtained from the flowers, leaves, and stems of the Lavender, Lavandula intermedia, Labiatae
Lavandula Intermedia Flower/Leaf/Stem Water	Lavandula Intermedia Flower/Leaf/Stem Water is an aqueous solution of the steam distillates obtained from the flowers, leaves, and stems of Lavandula intermedia, Lamiaceae
Lavandula Intermedia Oil	Lavandula Intermedia Oil is the volatile oil obtained from the whole plant of the Lavender, Lavandula intermedia, Labiatae

**Table 6 molecules-28-02986-t006:** Summary of products containing Lavandula Hybrida Oil (INCI) based on EWG data (US market only).

Product Type	Number of Products
Body wash/cleanser	59
Shampoo	43
Moisturizer	36
Facial moisturizer/treatment	33
Antiperspirant/deodorant	31
Facial cleanser	31
Conditioner	28
Bubble bath	24
Bath oil/salts	19
Serums/essences	19
Foundation	18
Liquid hand soap	18
Bar soap	16
Facial mask	15
Toners	13
Hair treatment/serum	11
Scrub/exfoliant	10
Baby soap	8
Baby shampoo	8
Sunscreen	7
Hand sanitizer	7
Body oil	5
Hand cream	4
Perfumes/Eau de toilette	4
Eye cream	3
Lip balm	3
Makeup remover	2
Other	29

## Data Availability

Not applicable.

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
