# Peer review of "Lavandula x intermedia—A Bastard Lavender or a Plant of Many Values? Part II. Biological Activities and Applications of Lavandin"

_molecules, 2023, doi:10.3390/molecules28072986_

Round 1

Reviewer 1 Report

The manuscript entitled “Lavandula x intermedia – A Bastard Lavender Or A Plant Of Many Values? Part II. Biological Activities And Applications Of Lavandin” aims to review all available original research articles on the biological effects of L. x intermedia essential oil and its other plant extracts. Furthermore, the authors reviewed the current applications of lavandin in industries and everyday life.

The manuscript represents part II in a series of articles, but part I is nowhere to be found. I think it should not be cited since there is no preprint even. Besides that, the manuscript is interesting and fits the scope of the journal. It is very informative and well-organized.

The abstract is suitable.

The introduction is very short and mainly describes the content of part I, so it should be improved.

The main part of the paper is good, very informative, and detailed.

The conclusion is sound.

The authors should check the numeration of sections. It looks like subtitle 2.1. is missing.

Author Response

We would like to thank the Reviewer for the careful and thorough reading of this manuscript and for constructive suggestions, which help to improve the quality of our manuscript. We have tried to do our best to respond to any comments. As indicated below, we have considered all the general and specific comments provided by the Reviewer and have incorporated changes in the text to reflect them.

The manuscript represents part II in a series of articles, but the part I is nowhere to be found. I think it should not be cited since there is no preprint even.

We sent both manuscripts (Part I and Part II) to the Molecules journal on the same day. We will provide the correct citation to the part I once it is accepted for publication. We asked editors to provide both parts for reviewers to assess.

The introduction is very short and mainly describes the content of part I, so it should be improved.

The introduction was revised and extended accordingly. We did not want to elaborate on the basic features of lavandin to avoid excessively repeating ourselves, as it was all covered in part I.

The authors should check the numeration of sections. It looks like subtitle 2.1. is missing.

Thank you so much for pointing it out. We have added some more new text and rearranged it to address this issue.

Reviewer 2 Report

The review article by Pokajewicz et al. is the second in a series aimed at providing an overview of chemical composition and biological activities of Lavandula x intermedia (lavandin).

The manuscript is well written but I suggest major revision. This literature review is too vague. The fact that the essential oils of lavandin possess some biological properties is obvious and known for a long time. But here is all described theoretically. In the paragraph 2.2.1 for example, the authors should mention MICs and MBCs or in any case experimental values rather than talk about moderate, weak or strong antibacterial activities of Eos. The same is for the paragraph 2.2.2. Indeed, this is a weak point in all the paragraphs in which the biological activity is described. The MS contains no detailed data on the mentioned effect of these essential oils in the medical area.

Moreover, the authors did not take into account that essential oil is not the only constituent of lavandin and completely ignored other compounds which may contribute significantly to the biological activity of this plant.

In addition, minor changes to the style and sentence structure are required:

Abstract, line 18 and below: the authors should clarify what kind of oil is this “Lavandin oil” (a fixed oil? An essential oil? They are completely different).

Pag 2 line 63: please update reference [3]

Pag 2 line 77: please change Table X with Table 1

Author Response

We would like to thank the Reviewer for the thorough reading of this manuscript and for constructive suggestions, which help to improve the quality of our manuscript. We have tried to do our best to respond to any comments. As indicated below, we have considered all the general and specific comments provided by the Reviewer and have incorporated changes in the text to reflect them.

The manuscript is well written but I suggest major revision. This literature review is too vague. The fact that the essential oils of lavandin possess some biological properties is obvious and known for a long time. But here is all described theoretically. In the paragraph 2.2.1 for example, the authors should mention MICs and MBCs or in any case experimental values rather than talk about moderate, weak or strong antibacterial activities of Eos. The same is for the paragraph 2.2.2. Indeed, this is a weak point in all the paragraphs in which the biological activity is described. The MS contains no detailed data on the mentioned effect of these essential oils in the medical area.

At first, we were surprised to see the opinion of the Reviewer that the literature review is too vague, as according to our best knowledge, we reviewed all original articles relating to Lavandula x intermedia and its biological effect. If the Reviewer knows about any missed original work, we would kindly ask for a suggestion. We will be more than happy to include it and increase the value of this manuscript.

We agree with the Reviewer that “lavandin possesses some biological properties is obvious and known for a long time”, we just wanted to stress the fact, that there is a significant discrepancy between its extensive use in the market and its limited scientific research. Your feedback has helped us to strengthen this point in our manuscript.

We would like to thank the Reviewer for suggesting that we provide more detailed information on the studies in the antimicrobial area. After considering the Reviewer’s suggestions, we see that we could have described some studies in more detail. We decided to view the literature again and provided much more info, where given also suggested MIC and MBC/MFC values. The same relates to Table 2 with other biocidal effects. The manuscript has been updated accordingly and is more informative now (it is now 37 pages in comparison to the original 26).

Regarding paragraph 2.2.2, we provided a short description of each study in Table 3 and have included a reference for readers who require more information and the exact results of different tests. Especially in the case of antioxidant power studies, where numerous methodologies exist (e.g. DPPH, ABTS, FRAP, ORAC, chelating power and reducing power, lipoxygenase inhibitory test), using different specific methods and different standard compounds, presenting the multiple values in the table would not add any value to the work an in our opinion it would be even confusing. Comparing the results of different authors in this field is very difficult and not always even possible. It can be easily observed in the discussion section of articles when often authors restrain from comparing their work to external results and build their discussion on the comparisons made within a set of their self-studied samples. Therefore, we believe that our decision to present a short description of each study and a reference for further information strikes an appropriate balance.

The MS contains no detailed data on the mentioned effect of these essential oils in the medical area.

As we mentioned earlier, we conducted a thorough review of all relevant original articles and included only studies with evidence-supported effects. Tables 1-3 in the manuscript indeed provide a comprehensive summary of the current picture of research in the area. Most studies are basic and performed on in vitro models. We were also surprised about the scarcity of research devoted to Lavandula x intermedia in the medical area as Lavandula angustifolia is much far more studied, including multiple clinical studies with Silexan, but this is reflective of the current state of research in this field.

Moreover, the authors did not take into account that essential oil is not the only constituent of lavandin and completely ignored other compounds which may contribute significantly to the biological activity of this plant.

It is indeed true that essential oil is not the only preparation of lavandin. In Part I of our manuscript, we discussed other phytochemicals detected in L. x intermedia (and only in this taxa). In Part II, we provided a detailed discussion of all the different preparations of lavandin that were investigated, including essential oils, hydrolates, aqueous, acetonic, and ethanolic extracts, etc. It is important to note that out of the 19 rows in Table 1, five describe studies of preparations of lavandin other than essential oil. Similarly, three out of twelve rows in Table 2, and six out of 17 rows in Table 3 also describe studies of these other preparations of lavandin. We would like to emphasize that the scarcity of research on the composition and effects of other lavandin preparations is not due to our intention or negligence, but rather reflects the current state of research in the literature.

Abstract, line 18 and below: the authors should clarify what kind of oil is this “Lavandin oil” (a fixed oil? An essential oil? They are completely different).

We have meant lavandin essential oil.  We have corrected the abstract and added relevant info in the introduction to clarify the meaning …” lavandin essential oil (further referred to as lavandin oil)…”.

 Pag 2 line 63: please update reference [3]

Reference updated.

 Pag 2 line 77: please change Table X with Table 1

Corrected

In case the reviewer has any additional specific comments on how we can further improve our manuscript or can suggest any original articles that could contribute to it, we would be more than happy to consider them. We greatly value the Reviewer's feedback and constructive criticism, and we believe that addressing the concerns and suggestions will help us produce a stronger manuscript that will benefit the scientific community. We thank Reviewer for her/his time and input.

Reviewer 3 Report

See all the remarks on the pdf file because special character for the hybrid botanical name of lavandin does not appear here.

Author Response

We would like to thank the Reviewer for the careful and thorough reading of this manuscript and for constructive suggestions, which help to improve the quality of our manuscript. We have tried to do our best to respond to any comments. As indicated below, we have considered all the general and specific comments provided by the Reviewer and have incorporated changes in the text to reflect them.

Remark for all the text: the botanical name is Lavandula ï‚´ intermedia Emeric ex Loisel. (see Upson & Andrews, 2004)

Thank you for bringing this to our attention. In the first part, we provided botanical characteristics, including taxonomy. However, it is important to also provide the correct botanical name in the second part. Therefore, we have presented the name in the introduction accordingly. Later in the text, we used interchangeable abbreviations and alternative names such as LI, L. x intermedia, lavandin, or Lavandula x intermedia.

Line 76 correction need for the number of the table X

Corrected

line 153: replace the references 2 and 4 that are reviews by those of the original papers.

Corrected

Line 290: add references about the cyclohexadiene-like components and antioxidant properties.

Citation inserted

Line 447: a paragraph about camphor is need because camphor is one of the main compounds of some lavandin cultivars. It has been well studied and some studies could be reported here. Its toxicity could explain the limited used of LI and LL for aromatherapy applications.

Thank you for this insightful suggestion as we had overlooked this aspect. We have added the discussion about camphor, and we believe it has significantly improved the quality of the article.

Line 564: add a reference about the idea of forage plant for pollinators.

Citation inserted

Round 2

Reviewer 2 Report

The manuscript can now be accepted for publication, the authors revised their manuscript accordingly with my comments